# Prediction of Antibody Viscosity from Dilute Solution Measurements

**DOI:** 10.3390/antib12040078

**Published:** 2023-12-01

**Authors:** Kamal Bhandari, Yangjie Wei, Brendan R. Amer, Emma M. Pelegri-O’Day, Joon Huh, Jeremy D. Schmit

**Affiliations:** 1Department of Physics, Kansas State University, Manhattan, KS 66506, USA; kamalb@ksu.edu; 2Amgen Inc., Thousand Oaks, CA 91320, USA; ywei04@amgen.com (Y.W.); bamer@amgen.com (B.R.A.); epelegri@amgen.com (E.M.P.-O.); jhhuh76@gmail.com (J.H.)

**Keywords:** viscosity, binding affinity, reptation, affinity-capture self-interaction nanoparticle spectroscopy (AC-SINS), dynamic light scattering (DLS)

## Abstract

The high antibody doses required to achieve a therapeutic effect often necessitate high-concentration products that can lead to challenging viscosity issues in production and delivery. Predicting antibody viscosity in early development can play a pivotal role in reducing late-stage development costs. In recent years, numerous efforts have been made to predict antibody viscosity through dilute solution measurements. A key finding is that the entanglement of long, flexible complexes contributes to the sharp rise in antibody viscosity at the required dosing. This entanglement model establishes a connection between the two-body binding affinity and the many-body viscosity. Exploiting this insight, this study connects dilute solution measurements of self-association to high-concentration viscosity profiles to quantify the relationship between these regimes. The resulting model has exhibited success in predicting viscosity at high concentrations (around 150 mg/mL) from dilute solution measurements, with only a few outliers remaining. Our physics-based approach provides an understanding of fundamental physics, interpretable connections to experimental data, the potential to extrapolate beyond training conditions, and the capacity to effectively explain the physical mechanics behind these outliers. Conducting hypothesis-driven experiments that specifically target the viscosity and relaxation mechanisms of outlier molecules may allow us to unravel the intricacies of their behavior and, in turn, enhance the performance of our model.

## 1. Introduction

Antibodies are promising therapeutics for a variety of ailments, but the large size of antibody molecules means that the total mass of protein that must often be administered to achieve a therapeutic effect can be very high. In most cases, the preferred method of delivery is subcutaneous injection, which imposes strict limitations on the delivery volume. At the high concentrations (e.g., ≥100 mg/mL) required to achieve injection volumes compatible with subcutaneous administration, many antibody solutions become too viscous for traditional subcutaneous injections or pose challenges for standard manufacturing operations [1,2,3,4]. In some cases, this high viscosity can be alleviated in formulation but, often, the only option is to reduce the protein concentration and switch to intravenous delivery, which incurs a much higher cost and less patient convenience than subcutaneous administration. Since formulation efforts to modify viscosity can be challenging and costly, it would be beneficial to select less viscous molecules in screening. However, viscosity problems are often not apparent until late in the development pipeline when it is possible to generate the high-concentration solutions necessary for rheological measurements. Therefore, there is a strong need for methods to predict viscosity behavior from the dilute solutions available in the early stages of development. However, such predictions are inherently difficult because the two-body interactions occurring in dilute solutions are fundamentally different from the many-body interactions that are responsible for the nonlinear increase in viscosity observed at high concentrations.

Many previous methods have been developed to predict antibody viscosity from molecular properties using database methods. One recent study [5] identified from a set of 59 therapeutic monoclonal antibodies (mAbs) that viscosity and opalescence could be predicted through measurement of weak self-interactions in dilute solution (second virial coefficient, *B_2_*, or dynamic light scattering parameter, *k_D_*). By contrast, in previous experimental work [6] using a similar size sample set, it was observed that mAb viscosities increase strongly with hydrophobicity and charge dipole distribution but decrease with net charge. Along these lines, computational models that employ spatial charge mapping (SCM) [7] calculate the charge on 3D homology models, resulting in an SCM score that can aid in identifying potentially viscous mAbs early in development. More recently, approaches that utilize machine learning have also been used to find the molecular properties responsible for high viscosity. One such study [8] found that the overall net charge and the amino acid composition in the mAb variable region (Fv) are key properties that influence viscosity. Combined, these approaches and others have identified key characteristics such as surface electrostatics, *k_D_*, *B_2_*, and hydrophobicity [4,6,7,9,10,11]. While powerful, these studies are often limited in scope and may use curated mAb sample sets, which can hamper broader predictive abilities.

Another approach to predicting protein viscosity is to mimic the many-body environment in silico. This approach seeks to model the various weak protein–protein interactions that ultimately lead to macroscopic viscosity challenges [12,13]. Due to mAb size and complexity as well as computational cost, such methods often require extensive coarse graining (CG) of the molecules to achieve the large systems needed to assess viscosity. In early studies, Li et al. [14] used a CG approach and observed that the net charge, ξ-potential, and pI of Fv regions correlated with high-viscosity mAb solutions. Similarly, Buck et al. [12] found that transient antibody networks are formed due to domain–domain electrostatic complementarities. They propose that these interactions are most likely the origin of high-concentration viscosity for mAb solutions. In work by Calero-Rubio et al. [15], they combined experimental light-scattering data to measure second osmotic virial coefficients (B_22_) in dilute conditions with CG modeling to predict high-concentration viscosity behavior. Interestingly, their studies revealed that the addition of sucrose, a common excipient in drug product development, induces stronger repulsions between protein molecules via the accumulation of sucrose around the protein surface. Key limitations of CG computational approaches are the poor representation of short-range interactions, the lack of model flexibility, and the lack of atomistic detail in CG modeling. Additionally, computational costs/complexity as well as expert user interpretation can limit the utility of this approach.

Physics-based models have also been applied as a method to understand antibody viscosity. In principle, these methods have broader applicability because the results can be extended to conditions outside the training set. They have also shown the potential to bridge disparate length scales. This is carried out by accounting for amino acid-scale differences between molecules via their effect on parameters within a solution model described by either reptation theory or Wertheim theory [16,17,18]. A key insight from these methods is that the primary driver of viscosity comes from the entanglement of elongated complexes, an observation that is consistent with the simulations of Buck et al. [12].

In this manuscript, we describe an approach in which physics-based theory is used to predict many-body solution characteristics from experiments that are sensitive to two-body interactions. We achieve predictive success comparable to other methods in the literature; however, because our approach is based on physical models, it provides testable explanations that allow for future refinement.

## 2. Materials and Methods

### 2.1. Reptation Model

Our theory for antibody viscosity is based on the observation that the differences between antibodies are mostly localized to the complementarity-determining region (CDR). Therefore, the interactions driving viscosity should originate from CDR-mediated association. This simple observation has important implications for the morphology of antibody clusters. This is because the two CDRs on each molecule limit the number of interaction partners a given molecule has within an antibody cluster. For example, if the antibodies interact via CDR-CDR interactions, the resulting clusters will be linear chains of antibodies. We refer to this as the head-to-head (HH) model [16]. Interactions between a CDR and another patch of the molecule increases the interaction valence of each molecule to three or four, which can result in branched chains. This is referred to as the head-to-tail (HT) model [17]. Importantly, in each case, the limited valence arising from CDR-mediated binding will favor the formation of elongated structures. This should be contrasted with compact aggregates, which require interactions that are not directionally specified (for example, in colloidal instability).

The average assembly size can be evaluated for both the HH and HT models. For the HT model, the result is [17]
(1)$${<ח>}_{HH}=\frac{2KC}{\sqrt{1+4KC}-1}$$
where *K* is the association equilibrium constant for the formation of antibody dimers by HH contacts, which is defined by $K=\frac{C_{2}}{{C_{1}}^{2}}$, where *C_2_* and *C_1_* are dimer and monomer concentration, respectively, and *C* is the total antibody concentration.

For the HT model, the average size is given by
(2)$${<ח>}_{HT}=\frac{\sqrt{2}SC}{\sqrt{1+4SC+S^{2}C^{2}-\left(1+SC\right)\sqrt{1+6SC+S^{2}C^{2}}}}$$
where *S* is the association constant for HT dimers. The HH and HT models are difficult to distinguish in the limits of low concentration and/or low affinity when branched structures are rare. Therefore, for the moment, we focus on the HH model and revisit this simplification later in the manuscript.

The existence of elongated structures has important consequences for the solution dynamics because they occupy a much larger effective volume than the substituent molecules and are prone to entanglement. Entanglement prevents elongated structures from diffusing laterally, so nearly all diffusion occurs through snake-like motion along the elongated direction [19]. This situation is described by reptation theory, which was originally conceived for long, flexible polymers but is also used to describe solutions of rodlike particles. In reptation theory, the solution viscosity is given by [16,17,19,20,21]
(3)$$ח\sim C^{\frac{3}{3\nu -1}}L^{3}$$
where *L* is the length of the elongated structures expressed in units of a monomer present at total concentration *C*. *ν* = 3/5 is the Flory exponent, which describes the effective volume occupied by the elongated structure.

Inserting our expression for the average complex size (Equation (1)) into Equation (3), we arrive at an expression for the viscosity of an antibody solution.
(4)$$ח=AC^{\frac{3}{3\nu -1}}{\left(\frac{2KC}{\sqrt{1+4KC}-1}\right)}^{3}$$
where *A* is a proportionality constant with the value 5.4×10^−8^cP(mg/mL)^3.75^ [16]. This constant describes lateral interactions between elongated antibody complexes. Therefore, it will be constant within a given antibody scaffold, but it can change due to disulfides within the hinge region that modify molecular flexibility or under solution conditions that promote non-specific interactions.

Equation (4) describes complex, many-body interactions driving antibody viscosity in terms of a parameter, *K*, that describes two-body interactions. This means that if *K* can be measured through dilute solution methods, we can predict the viscosity at high concentration. However, Equation (4) relies on several assumptions including the following: (1) HT interactions are ignored, (2) the dimerization physics at high concentration is approximately the same as that at low concentration, and (3) the “slithering” dynamics captured in the “*A*” parameter are constant. We expect that each of these assumptions will be violated in certain cases, but these cases can be experimentally identified and used to further refine the method.

### 2.2. Experimental Methods

A data set of 89 mAbs was generated by Amgen [22]. This data set consists of AC-SINS plasmon shift (Δ*λ*), AC-SINS diffusion coefficient (*D_np_*), dynamic light scattering interaction parameter (*k_D_*), viscosity at _~_70 mg/mL and _~_150 mg/mL at 25 °C. Mock et al. [22] describes the experimental procedures to generate the data set including buffer conditions, coupling protocol to the nanoparticles, etc. Out of these 89 mAbs, 63 molecules were randomly selected as a training set, 17 molecules were randomly selected as a validation set, and the remaining 9 molecules were used as a test set. By fitting Equation (4) to these viscosity values, we obtained dimerization affinity parameter, *K*, for each molecule in the sample set.

We employed two methods to assess the two-body interaction parameter, *K*.

AC-SINS (Affinity-Capture Self-Interaction Nanoparticle Spectroscopy) is an assay in which antibodies are attached to gold nanoparticles [23,24,25,26]. The resulting nanoparticles will then associate via antibody–antibody (Ab-Ab) interactions, which can be detected as either a reduction in the nanoparticle diffusivity or a shift in the plasmon wavelength. AC-SINS has advantages where, by clustering antibodies on the surface of the nanoparticle, it mimics the high-concentration environment and provides an amplification effect to increase the sensitivity to weak interactions. However, the method by which antibodies are tethered to the nanoparticle will prevent certain regions of the antibody from interacting with other antibodies. Therefore, we expect that AC-SINS will be insensitive to head-to-tail interactions. In addition, after the antibody capture protocol, there remains a significant concentration of soluble test antibody present in solution. This may result in three-body “bridging” interactions that complicate interpretation.

DLS (dynamic light scattering) can be used to measure the concentration-dependent change in antibody diffusivity [27,28]. Attractive interactions will result in a decrease in diffusion while repulsive interactions will increase diffusion. Since the antibodies are untethered in this experiment, DLS is sensitive to the HT interactions that are missing in AC-SINS. The observed diffusivity change is the result of a statistical average of all interactions. This can also complicate analysis if repulsive interactions between certain parts of the molecule (which minimally contribute to the viscosity profile) prevent the formation of attractive interactions between regions of the molecule that strongly influence viscosity.

Despite these limitations, we expect that both AC-SINS and DLS measurement will be most strongly affected by the CDR-dependent interactions that drive viscosity.

### 2.3. Description of Algorithm

Equation (4) describes the concentration-dependent viscosity of an antibody solution as a function of a single parameter, *K*, which describes the first dimerization event in the formation of elongated antibody complexes. Thus, we can predict the high-concentration viscosity if we can estimate *K* using methods such as those described in the previous section.

The major pitfall to this approach is that it implicitly assumes that the variation in dilute solution interactions is dominated by a single binding mode that is also responsible for the formation of elongated complexes at high concentration. The presence of multiple interaction modes will complicate the correlation between the dilute properties and *K*, particularly if there is an interaction that results in the formation of compact assemblies, which will minimally contribute to the viscosity. Another complication is that some protein–protein interactions, such as electrostatic interactions [29,30,31], can have many-body contributions, causing them to qualitatively differ between the dense and dilute environments. Additionally, Equation (4) assumes that the major relaxation mechanism in the concentrated state is the reptation diffusion of antibody complexes. However, if the complexes have a branched structure, rather than linear, then reptation will be strongly suppressed. As a result, antibodies that interact head-to-tail will have qualitatively different behaviors than head-to-head antibodies above the concentration where branching becomes significant [17].

The effect of these complicating factors can be seen in Figure 1, which shows the correlation between the diffusion of AC-SINS nanoparticles, the *k_D_* parameter, and the *K* values obtained from fitting the viscosity profile to Equation (4). Some of the *K* values in Figure 1 are very small, and in a few cases (training set molecule ID 17 (Tr_17_), Tr_26_, Tr_46_, and Tr_51_) are even negative. These non-physical values are indicative that these molecules are not experiencing entangled dynamics and, therefore, Equation (4) is not applicable. Instead, a non-entangled theory accounting for colloidal collisions should be used [17]. These four molecules are excluded from our fits. Both *D_np_* and *k_D_* show a strong correlation between the dilute measurements and *K*, along with a significant number of outliers. Upon closer inspection, many of these outliers can be understood. Molecules above the trend line in Figure 1A,B are those whose viscosity is greater than what is expected from the AC-SINS experiment. This behavior is consistent with HT binding between antibodies, which will cause the AC-SINS experiment to predict anomalously low viscosity for two reasons. The first reason is that one of the interaction sites responsible for the formation of elongated complexes may be sterically inhibited by the tether of the antibody to the nanoparticle. This explanation is supported by the fact that many of these molecules fall near the trend line in the DLS experiment (Figure 1C) in which the entire molecule is accessible for interactions. Further support comes from the fact that these molecules also deviate from the *D_np_* vs. *k_D_* trend line, consistent with the nanoparticle interfering with attracting interaction (Figure 2). The second reason is that strong HT binding will lead to the formation of branched antibody complexes [17] that are unable to relax through the reptation mechanism described by Equation (4) and, therefore, will have much a higher viscosity than would be expected from linear complexes. This is consistent with the fact that highly viscous molecules appear in both the AC-SINS and DLS experiments (Figure 1A,C).

Further inspection of Figure 1A shows that the correlation between viscosity and nanoparticle diffusion breaks down for diffusion rates less than 1 μm^2^/s. It is likely that this is because the Stokes radius $R_{S}\propto N^{\frac{1}{d_{f}}}$ grows less strongly as the number of particles in a cluster, *N*, grows larger. $d_{f}$ is the fractal dimension of a cluster of nanoparticles, which will lie between the values $d_{f}=3$ for compact clusters and $d_{f}=1.8$ for diffusion-limited aggregation [32,33]. Therefore, nanoparticle diffusion is expected to have poor sensitivity for strongly interacting antibodies, due to the large clusters found in these cases.

To avoid the poor sensitivity in the strong interaction regime, we limit our analysis to antibodies with *D_np_* > 1 μm^2^/s. We also exclude the three molecules identified as suspected HT binders. The best fit line for the remaining molecules is
(5)$$D_{np}=4.9-2045\;K\;\;\;\;$$

For the DLS experiments, we omit two sets of outliers before obtaining the best-fit line. The first is a single molecule with a *k_D_* = 84 mL/g, which is inconsistent with the expected range for antibodies, suggesting an experimental error [3,34]. The second set is molecules with *K* > 0.01 mL/mg, which, as we explain below, are unlikely to be described by Equation (4). The greater number of molecules included in the *k_D_* fit is because we expect that *k_D_* is sensitive in cases where the nanoparticle will distort the results. The best fit line for the included molecules is
(6)$$k_{D}=\;56.03-\;15,000\;K$$

To summarize, our algorithm predicts the viscosity using either *D_np_* or *k_D_* values measured in dilute solution. These values can be input into the best fit lines given by Equations (5) or (6) to estimate the association parameter, *K*, from AC-SINS and DLS experiments, respectively. The *K* parameter can then be used in Equation (4) to predict the viscosity at the high concentrations where antibodies are in the entangled regime. This prediction will be most reliable for weak/moderate interacting antibodies that lie within the sensitive range of the AC-SINS experiment.

## 3. Results and Discussions

Figure 3 compares the predicted viscosity curves for the validation set of 17 molecules to the measured viscosity at two concentrations. The correlation between the predicted and measured viscosities is comparable to the training set (Figure 1). Specifically, there are two molecules (validation set molecule ID 1 (V_1_) and V_15_) in which the DLS prediction is good, but the nanoparticle diffusion (*D_np_*) under-predicts the viscosity. Since this is the expected result for HT binders, we predict that structural analysis would reveal an interaction between the CDR and a location occluded by nanoparticle and/or capture antibody. There are another three molecules (V_2_, V_7_, and V_9_) in which both methods under-predict the viscosity. These molecules lie below the 1 μm^2^/s threshold where the correlation between nanoparticle diffusion (*D_np_*) and viscosity is observed to break down (Figure 1). More experiments will be required to determine if this breakdown can be rectified by refining the experimental protocol to account for the long equilibration times expected of strongly binding molecules or if it is indicative of a different physical regime (for example, reptative and non-reptative relaxation for HH and HT binders, respectively).

We further evaluate the performance of our model using test data of nine additional molecules. Both ACSINS and DLS experiments measure the viscosity at 4 or 5 discrete point concentrations ranging from 70 mg/mL to 250 mg/mL, which allow us to evaluate the accuracy of the model-predicted concentration–viscosity curve.

In the log–log representation of the actual data (Figure 4), it becomes evident that viscosities above 100 mg/mL and below 100 mg/mL manifest distinct power regimes, a phenomenon previously documented by Schmit et al. [16]. The observed poor agreement at concentrations below 100 mg/mL is attributed to the molecules being excessively dilute, thus failing to exhibit the entangled dynamics described by Equation (4). Overall experimental viscosities agree with the predicted viscosity curve shown in Figure 4. For test set molecules ID 2 (Te_2_) and ID 4 (Te_4_), both DLS and ACSINS algorithms underpredict the viscosity. As we observed in the training and validation sets, both outliers have *D_np_* less than 1 μm^2^/s.

Our model highlights the need to better understand the physics of antibody viscosity and effects that lead to the scatter in our fits (Figure 1). Of these effects, the most important is head-to-tail binding. HT binding is difficult to assess through AC-SINS because the nanoparticle and/or capture antibody can sterically inhibit, or even occlude the binding sites responsible for increasing viscosity. This difficulty mitigates the sensitivity advantage of AC-SINS over DLS. While the occlusion of binding sites could be avoided using different ligation techniques, a larger uncertainty comes from the lack of reptation as a relaxation mechanism for branched antibody complexes. Resolving this uncertainty will require a comparison of viscosity curves of molecules for which the intermolecular association sites have been identified through techniques such as Hydrogen–Deuterium Exchange [35,36]. Another possible source of outliers is antibody association sites that are not compatible with the formation of elongated structures. Such a site would contribute to lower diffusion in AC-SINS and DLS experiments but would not lead to the entanglements that are the primary contributor to increased viscosity. Finally, a central weakness of dilute solution measurements in predicting high-concentration behavior is the possibility that many-body effects can invalidate the assumption of pairwise interactions. Such non-pairwise interactions are especially prevalent for charged molecules at high concentration [31,37]. The dense layer of proteins at the surface of AC-SINS nanoparticles may capture some many-body effects, although the geometry dependence of electrostatic screening interactions [29] makes it unclear how much the nanoparticle surface can recapitulate a high-concentration solution.

To gain further insight into the relationship between *D_np_*, *k_D_*, and viscosity, we generated a 3D plot of all molecules in the training set (Figure 5). Inspection of this plot suggests that our proposed cutoff for HT binders (Figure 2) may have been too conservative. A revised cutoff line (black dashed line) divides the set into a region above the line where all three quantities are well correlated and a region below the line containing three sets of outliers. The first set of outliers is the previously identified HT molecules (red in Figure 2 and Figure 5). The second set is the high-viscosity outliers that extend off the scale of the plot. These unphysically large affinity constants are likely to be the result of fitting large, branched HT complexes, which cannot undergo reptation, to the reptation model. Third, the revised cutoff line separates a cluster of molecules at about *k_D_* = 20 mL/g that deviates from the main group. Again, these molecules have a *D_np_* greater than we would expect for the observed *k_D_* and viscosity. But the overall affinity is too weak for these to show up as outliers in our original analysis.

Next, we evaluate whether this model can be used to assist the de-selection of viscous antibody molecules at the early candidate screening stage. In Table 1, both experimental and predicted viscosity values (based on *D_np_* and *k_D_*) at ~150 mg/mL and experimental viscosity are listed for the 17 test molecules.

Here, we propose a binary classification system to categorize these mAbs using a cutoff value of 20 cP. Color coding was then applied to differentiate high- (>20 cP) vs. low- (≤20 cP) viscosity molecules as measured at 150 mg/mL and experimental concentrations. This cutoff is somewhat complicated by the fact that the experimental concentration often deviated from this value, which is significant for some borderline cases. Out of the 17 test molecules in Table 1, the viscosity models flagged 3 molecules (V_1_, V_2_, and V_15_), all 3 of which were experimentally found to be viscous. Three additional molecules (V_7_, V_9_, and V_11_) were experimentally found to lie above the viscosity threshold. In the case of molecule V_11_, the predicted viscosity is actually very close to the actual viscosity if we evaluate the viscosity at the same concentration of the experimental measurement. The remaining two mAbs (V_7_ and V_9_) are the two identified as outliers in (Figure 3 and related discussion).

The same comparison was then conducted on a test set of nine molecules in Table 2. Since viscosity was collected at multiple concentrations for these samples, the viscosity concentration curve was derived. Viscosities at exactly 150 mg/mL were calculated and compared with their corresponding predictions. Predictions suggest that all molecules have low viscosity, and 7 of the 9 molecules did have low experimental viscosity. However, two molecules (Te_2_ and Te_4_) were found to have high viscosity. This again is because the model has limited predictive power for the high-viscosity regime. We have incorporated a correlation graph into the Appendix A to provide a more comprehensive evaluation of both the validation data set (Table 1) and the test data set (Table 2). Overall, we can conclude that (1) the model can effectively flag the viscous molecules and (2) the model still faces challenge with predictions for viscosity above 40 cP.

## 4. Conclusions

Our method for viscosity prediction has the greatest predictive power for antibodies in the intermediate-viscosity regime. This regime has, perhaps, the greatest practical importance since high-viscosity molecules are more easily excluded through other methods. The performance of our method in the intermediate regime is a direct result of several choices in our model development. First, our method is based on a reptation model that is most accurate when the antibody complexes are large enough to behave like an elongated object, yet short enough that it is reasonable to neglect branching, complex fragmentation, and other effects that promote non-reptative relaxation mechanisms. To further target the intermediate regime, we have used the diffusion rate of AC-SINS nanoparticles because it is more sensitive to moderate interaction affinities, whereas the AC-SINS plasmon shift is more sensitive to stronger interactions. Finally, in fitting the correlation parameters for our model, we have only used training set data that fall within the sensitivity ranges of both the physical model and the AC-SINS diffusion. Given these choices, it is not surprising that the poorly predicted outliers are predominately located in the high-viscosity regime, which was not considered in the development of our model. Our physics-based approach has several advantages compared to empirical or machine learning methods. First, our model provides the functional form of the viscosity curve, which allows us to make viscosity predictions at different concentrations. Second, the physics-based model has known approximations and assumptions that provide testable hypotheses for both outliers and the failure of the model in the strong interaction regime (physical models of the weakly interacting regime have already been developed) [17]. Thus, the performance of the model can be improved through hypothesis-driven experiments that probe the viscosity and relaxation mechanism of outlier molecules. While improvements in model performance with additional data are a common feature of any predictive method, the physics-based approach has the additional advantage of providing intuition to guide sequence modification and formulation strategies.

## Figures and Tables

**Figure 1 antibodies-12-00078-f001:**
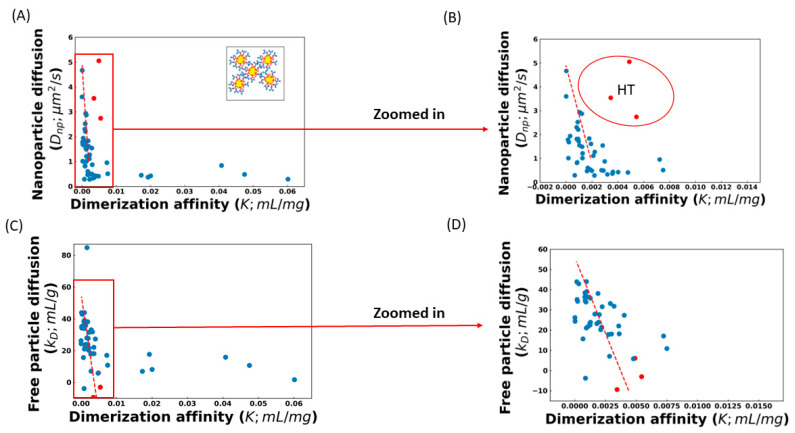
Dilute solution measurements of Ab-Ab association are correlated with the dimerization constant obtained from fitting viscosity profiles. Larger dimerization constants are equivalent to higher viscosity. (**A**) AC-SINS nanoparticle diffusion (*D_np_*) decreases with stronger dimerization due to the formation of nanoparticle clusters. (**B**) Zoomed view of the low-affinity regime. Red fit line represents the best fit line and we have excluded all HT outliers and molecules with *D_np_* less than 1 μm^2^/s. Panels A and B both show a population of antibodies significantly above the trend line, indicating that the viscosity is higher than predicted by nanoparticle diffusion. These outliers are consistent with head-to-tail binding. (**C**) Comparison of the *k_D_* parameter measured through DLS to the dimerization parameter, and a zoom in of the low-affinity regime (**D**). Red dashed line represents the best fit line. Again, stronger dimerization affinity is correlated with lower *k_D_*. The fewer outliers are consistent with the higher accessibility of binding sites in the absence of the nanoparticles. In all panels, red dots are the suspected HT binders, while blue dots are the remaining molecules.

**Figure 2 antibodies-12-00078-f002:**
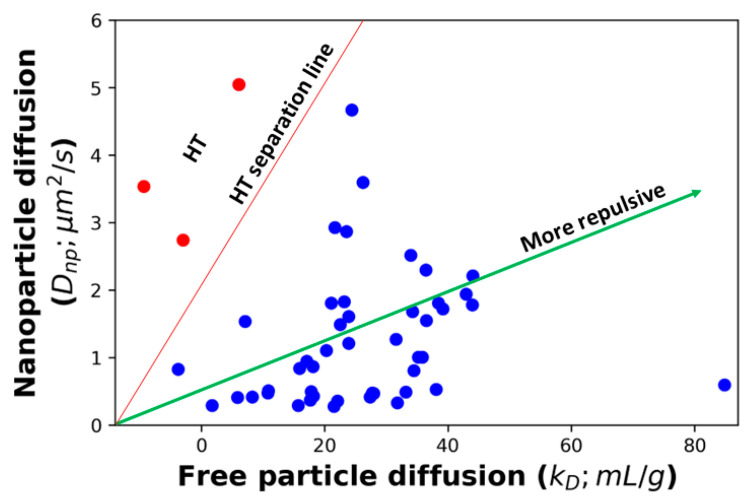
There is a strong correlation between nanoparticle diffusion and free particle diffusion measurements, although there are some HT outliers. HT separation line separates the HT molecules from all other molecules. Red dots are the suspected HT binders, while blue dots are the remaining molecules.

**Figure 3 antibodies-12-00078-f003:**
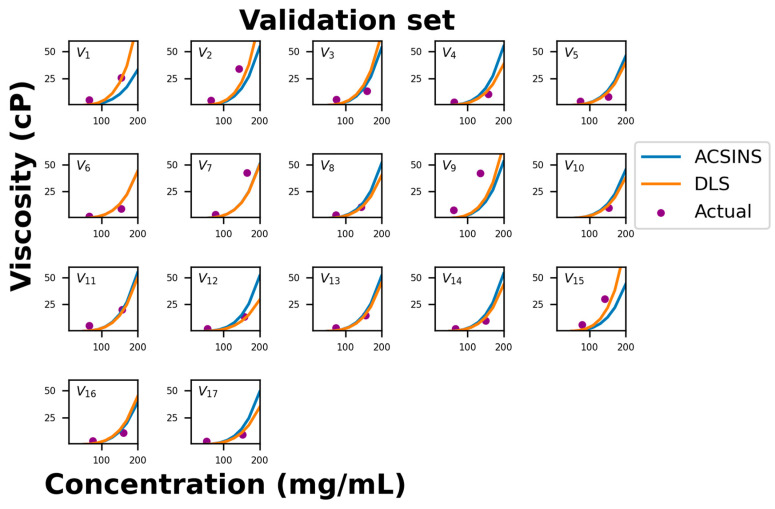
Comparison of experimental viscosity (purple dots) vs. predicted viscosities from *D_np_* (blue line) and *k_D_* (orange line) for the validation data set. There are two HT outliers (molecules V_1_ and V_15_) and three highly viscous outliers (molecules V_2_, V_7_, and V_9_).

**Figure 4 antibodies-12-00078-f004:**
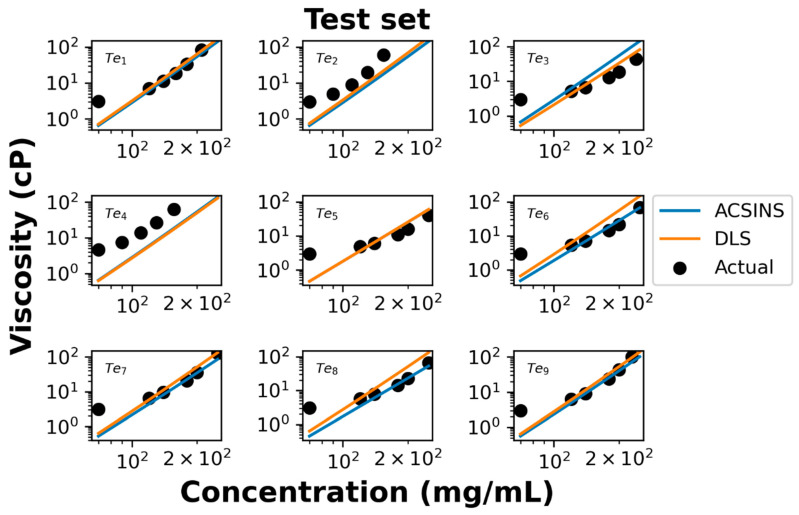
Comparison between experimental viscosity (black dots) with predicted viscosities from *D_np_* (blue line) and *k_D_* (orange line) for the test data set. The poor agreement at concentrations below 100 mg/mL is because the molecules are too dilute to exhibit the entangled dynamics described by Equation (4). The poorly predicted molecules (test set molecule ID 2 (Te_2_) and Te_4_) both fall below the 1 μm^2^/s threshold where the correlation between *D_np_* and viscosity breaks down.

**Figure 5 antibodies-12-00078-f005:**
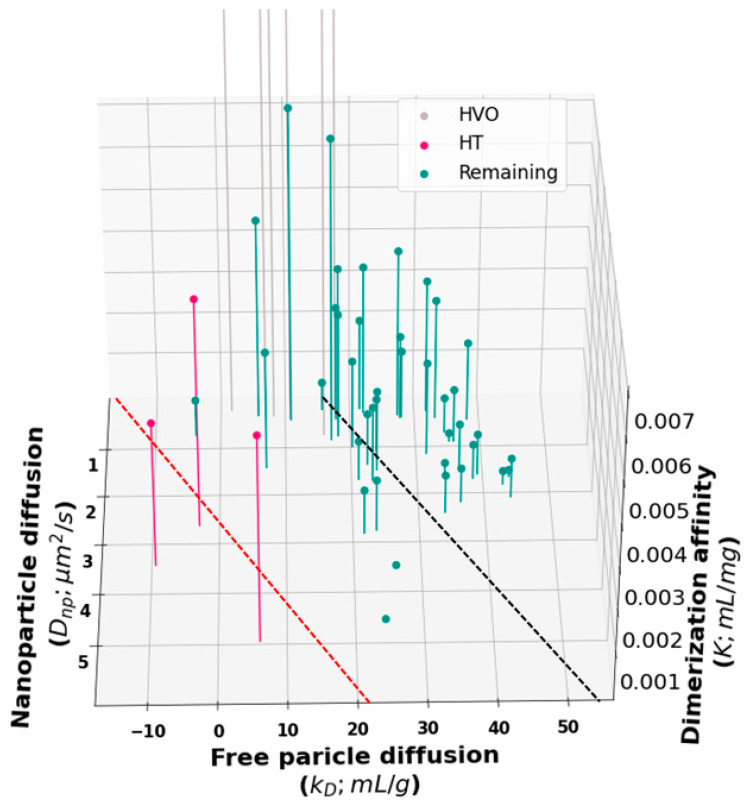
Three-dimensional representation of the data in Figure 1 and Figure 2 showing the nanoparticle diffusion (*D_np_*), free particle interaction parameter (*k_D_*), and dimerization affinity (*K*) for all molecules in the training set. The faint gray lines represent highly viscous outliers (HVOs), the vertical red lines represent previously identified HT molecules in Figure 1 and Figure 2, and green lines represent the remaining molecules in the training set. The red dashed line separates the previously identified HT outliers. The black dashed line is a proposed new cut-off line for the identification of HT molecules.

**Table 1 antibodies-12-00078-t001:** Our reptation model has a strong performance in predicting viscosity of molecules in validation data set except some outliers. Our model also classifies them in the outlier groups. Molecules having viscosity less than or equal to 20 cP (green box) are considered good molecules and those above 20 cP (red box) are bad molecules in the early-stage molecule selection for drug development pipeline. In terms of performance, “False positive” means both AC-SINS and DLS predictions suggest low viscosity, whereas experimental data show high viscosity. “False negative” means that either AC-SINS or DLS prediction suggests high viscosity, whereas experimental data show low viscosity. Other scenarios are considered as “True”.

				Validation Data Set				
Molecule ID	Experimental Concentration (mg/mL)	Experimental Viscosity (cP)	Viscosity (cP) at Exp. Concentration			Viscosity (cP) @ 150 mg/mL		
			AC-SINS Prediction	DLS Prediction	Performance	AC-SINS Prediction	DLS Prediction	Performance
1	154	25.9	11.5	23.6	TRUE	10.4	21.0	TRUE
2	143	34.2	12.8	17.5	False positive	15.7	21.6	TRUE
3	159	13.4	20.0	24.1	False negative	15.6	18.7	TRUE
4	158	10.6	19.9	14.5	TRUE	15.9	11.7	TRUE
5	152	8.1	14.3	12.7	TRUE	13.6	12.0	TRUE
6	154	9.1	14.6	14.6	TRUE	13.1	13.1	TRUE
7	165	42.5	22.1	21.8	TRUE	14.7	14.5	False positive
8	144	10.5	12.6	10.2	TRUE	15.0	12.1	TRUE
9	135	42.1	9.8	12.1	False positive	15.4	19.1	False positive
10	153	9.9	14.5	12.6	TRUE	13.4	11.6	TRUE
11	157	20.1	19.2	17.7	False positive	15.8	14.6	False positive
12	157	13.8	18.2	11.3	TRUE	15.0	9.4	TRUE
13	156	15	17.8	15.9	TRUE	15.1	13.5	TRUE
14	150	10	15.6	13.0	TRUE	15.6	13.0	TRUE
15	142	30.1	10.4	17.1	False positive	13.1	21.8	TRUE
16	160	10.9	15.5	17.5	TRUE	11.9	13.3	TRUE
17	152	9.1	15.2	11.4	TRUE	14.4	10.8	TRUE

**Table 2 antibodies-12-00078-t002:** Our reptation model also has a strong performance in predicting viscosity of molecules in test data set. Concentrations used for experimental measurement deviated somewhat 150 mg/mL depending on antibody, please see reference [22] for the precise value. There are two outliers (Te_2_ and Te_4_), which are also outliers in our model classification. Molecules having viscosity less than or equal to 20 cP (green box) are considered good molecules and those above 20 cP (red box) are bad molecules in the early-stage molecule selection for drug development pipeline. In terms of performance, “False positive” means that both AC-SINS and DLS predictions suggest low viscosity, whereas experimental data show high viscosity. “False negative” means that AC-SINS or DLS prediction suggests high viscosity, whereas experimental data show low viscosity. Other scenarios are considered as “True”.

		Test Data Set		
Molecules ID	Viscosity (cP) @ 150 mg/mL			Performance
	Experimental	ACSINS Prediction	DLS Prediction	
Te_1_	16.5	15.5	17.8	TRUE
Te_2_	44	15.6	19.3	False positive
Te_3_	8.8	15.8	10.4	TRUE
Te_4_	49.3	15.2	14.4	False positive
Te_5_	7.9	8.3	8.4	TRUE
Te_6_	9.7	9.0	15.8	TRUE
Te_7_	13.1	10.7	14.7	TRUE
Te_8_	10.1	7.8	14.8	TRUE
Te_9_	13.9	12.1	15.2	TRUE

## Data Availability

The data used to support the findings of this study can be made available by the corresponding author upon request.

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
