# Peer review of "Prediction of Antibody Viscosity from Dilute Solution Measurements"

_2073-4468, 2023, doi:10.3390/antib12040078_

Round 1
Reviewer 1 Report
Comments and Suggestions for Authors
The paper proposes the development of a tool to assess the viscosity of protein samples, in particular antibodies, of therapeutic interest. This is not necessarily something innovative, as there are other similar methods already described in the literature. However, the lack of improved tools with experimental validation that can be used for this purpose justifies its publication. The methodology's design is consistent and has been experimentally validated.
Author Response
Thank you for your review and valuable feedback on our manuscript. We appreciate your recognition of the importance of our proposed viscosity assessment tool, given the lack of improved tools with experimental validation.
Reviewer 2 Report
Comments and Suggestions for Authors
The manuscript by Bhandari et al. have developed a approach to predict the antibody viscosity at higher concentration. The manuscript is well written and clearly highlights the application and limitation of work. One question authors should be discuss in the manuscript would be "what are the advantages of author’s approach instead of in vitro experiments using higher Ab concentration since it would still need in vitro experiments with diluted concentration.
Minor comments:
1. The statement in introduction “large size of antibody means that a large mass must often be administered” is not correct. The antibodies are “relatively large to peptides” but still in nanometer scale. It is more related to bioavailability, effective neutralization etc.
2. A suggestion for the abstract and introduction section, the terminology can be changed for effective communication to the readers.
the large mass ïƒ high antibody doses; therapeutic doses ïƒ therapeutic effect
for example: "that a large mass must often be administered to achieve a therapeutic dose" can be "that high antibody doses must often be administered to achieve a therapeutic effect (* leading to further discussion on requirement of higher concentration)". Similar changes can be made in first line of abstract.
3. “engineer the primary sequence of molecules that may have viscosity issues” correction: “will not have viscosity issues”.
4. A table for the training dataset can also be provided for the readers who want to test/compare the current study.
Author Response
Thank you sincerely for your thoughtful and constructive feedback on our manuscript. We appreciate the time and effort you invested in reviewing our work. We carefully considered your comments and have made the following revisions to address your concerns.
Question/suggestion1: What are the advantages of author’s approach instead of in vitro experiments using higher Ab concentration since it would still need in vitro experiments with diluted concentration?
Answer: Material availability is limited at early candidate screening stage, so directly measuring viscosity of candidates at high concentrations is NOT feasible and/or costly. Our predictive tool enables rapid screening of candidate using very low amount of material.
Response: I have highlighted these sentences in our manuscript. “Since formulation efforts to modify viscosity can be challenging and costly, it would be beneficial to select less viscous molecules in screening or engineer the primary sequence of molecules to reduce the viscosity issues. However, viscosity problems are often not apparent until late in the development pipeline when it is possible to generate the high concentration solutions necessary for rheological measurements. Therefore, there is a strong need for methods to predict viscosity behavior from the dilute solutions available in the early stages of development.”
Minor Comment1: The statement in introduction “large size of antibody means that a large mass must often be administered” is not correct. The antibodies are “relatively large to peptides” but still in nanometer scale. It is more related to bioavailability, effective neutralization etc.
Answer: Specifically, our reference to the "large size of the antibody" in this context pertains to the total quantity required rather than the physical dimensions of the antibody itself. We have modified the sentences based on the reviewer's suggestions.
Response: The response to comment2 navigates the concerns raised in both comment1 and comment2.
Minor Comment2: A suggestion for the abstract and introduction section, the terminology can be changed for effective communication to the readers. the large mass à high antibody doses; Therapeutic doses à therapeutic effect.
Answer: Incorporating your insightful suggestions, we have refined several sentences to enhance clarity and precision.
Response: 1) Abstract first line: Often the high doses of antibody required for therapeutic effect necessitate high concentration products that can lead to challenging viscosity issues in production and delivery. 2) Introduction first line: Antibodies are promising therapeutics for a variety of ailments, but the large size of antibody molecules means that the total mass of protein that must often be administered to achieve a therapeutic effect can be very high.
Minor Comment3: “engineer the primary sequence of molecules that may have viscosity issues” correction: “will not have viscosity issues”.
Answer: We have corrected the sentence.
Response: It would be beneficial to select less viscous molecules in screening.
Minor Comment4: A table for the training dataset can also be provided for the readers who want to test/compare the current study.
Answer: We have referenced a recently published paper [References:22] from the AMGEN research group, wherein some of the authors are co-authors of our current paper for the dataset used in our study.
Response: I have highlighted this sentence in our manuscript. “A data set of 89 mAbs was generated by Amgen [22].”
Reviewer 3 Report
Comments and Suggestions for Authors
This work assesses a prediction model for high viscosity in high-concentration antibody formulations based on a previously published physical model describing entanglement of elongated complexes. Predictions models are for relevant for the field of therapeutic antibody development since material is limited in the discovery phase.
Following improvements/ unclarities should be addressed:
1. Method description is incomplete: add instruments used, if applicable parameter sets, buffer conditions of the antibodies used (with/without salt...), coupling protocol to the nanoparticles. Description of how the K-value was derived from kd/Dnp and afterwards, from the K-value the viscosity at a given concentration (e.g. one example to better follow the calculations?)
2. Figure 4: It seems that the actual data systematically follow an exponential behavoir rather than the flat curves predicted by ACSINS and DLS. Is a comment on this possible?
3. Figure 5: the 3D plot is difficult to understand when printed on 2D paper. Try to improve the results presentation (additional 2D sections?)
4. Table 1 and 2: the experimental viscosity is given with one significant number after the decimal point, whereas the prediction values are given with two significant numbers. Is this realistic? What is the estimated error of prediction? Is it possible to represent the data in Table 1 and 2 as correlation graph for better assessment? Notably, the experimental viscosity range is much larger (8.1-42.5) compared to the predicted (10-22 for AC-SINS and 10-24 for DLS). As a consequence, the cut-off value for "bad"/"good" molecules (20 cP) is very clear for experimental data but there are quite a few molecules slightly below or above 20 cp in the prediction models and it really depends on the error of prediction (e.g. is 19.95 really below 20 and the molecule is "good"??). What is the reason for the seemingly lower dynamic range of the prediction model?
5. Comparison with other prediction methods: Did you perform a comparison of the prediction of the here presented model to other models described in the introduction?
6. Application of the prediction: as stated in the conclusion the strength of the model is prediction in the intermediate viscosity regime. What would therefore be the recommendation in terms of selection of an antibody candidate after applying the prediction model?
7. Measurements were performed with concentrations of ~70 and ~150 mg/mL. However, the title of the publication draft announces "prediction of antibody viscosity from dilute solutions". Are there experiments available showing prediction from diluted solutions?
Author Response
Thank you sincerely for your thoughtful and constructive feedback on our manuscript. Your time and effort in reviewing our work are truly appreciated. Having carefully considered your comments, we have implemented the following revisions to address your valuable concerns.
Question1: Method description is incomplete: add instruments used, if applicable parameter sets, buffer conditions of the antibodies used (with/without salt...), coupling protocol to the nanoparticles. Description of how the K-value was derived from kd/Dnp and afterwards, from the K-value the viscosity at a given concentration (e.g. one example to better follow the calculations?)
Answer: Please see the comment above to referee 2 regarding a recent publication from the AMGEN research group for the complete method description and tabulated data. Also, we have a section (section 2.3: Description of algorithm) to elucidate the process of deriving the K-value from kD and Dnp and obtaining viscosity from the derived K-value.
Response: We have added and highlighted the relevant sentences based on reviewer’s suggestion. “A data set of 89 mAbs was generated by Amgen [22]. This data set consists of AC-SINS plasmon shift (Δλ), AC-SINS diffusion coefficient (Dnp), dynamic light scattering interaction parameter (kD), viscosity at ~70 mg/mL and ~150 mg/mL at 25 °C. Mock et al [22] describes the experimental procedures to generate the dataset including buffer conditions, coupling protocol to the nanoparticles etc.”
“To summarize, our algorithm predicts the viscosity using either Dnp or kD values measured in dilute solution. These values can put input into the best fit lines given by Eqs. 5 or 6 to estimate the association parameter K, from AC-SINS and DLS experiments, respectively. The K parameter can then be used in Equation 4 to predict the viscosity at the high concentrations where antibodies are in the entangled regime. This prediction will be most reliable for weak/moderate interacting antibodies that lie within the sensitive range of the AC-SINS experiment.”
Question2: Figure 4: It seems that the actual data systematically follow an exponential behavior rather than the flat curves predicted by ACSINS and DLS. Is a comment on this possible?
Answer: Figure 4 depicts a log-log plot for both actual and predicted data. In the log-log representation of the actual data, it becomes apparent that viscosities above 100 mg/mL and below 100 mg/mL exhibit distinct power regimes in relation to concentration, a phenomenon previously documented by Schmit et al in 2014 (reference 16, Figure 4). Notably, our entanglement model demonstrates accurate predictions within the intermediate concentration regime. The poor agreement at concentration below 100 mg/mL is because the molecules are too dilute to exhibit the entangled dynamics.
Response: We have added some sentences based on reviewer’s concern. “In the log-log representation of the actual data (Figure 4), it becomes evident that viscosities above 100 mg/mL and below 100 mg/mL manifest distinct power regimes, a phenomenon previously documented by Schmit et al [16]. The observed poor agreement at concentrations below 100 mg/mL is attributed to the molecules being excessively dilute, thus failing to exhibit the entangled dynamics described by Equation 4. Overall experimental viscosities agree with the predicted viscosity curve shown in Figure 4. For test set molecules ID 2 (Te2) and ID 4 (Te4), both DLS and ACSINS algorithm underpredict the viscosity. As we observed in the training and validation sets, both outliers have Dnp less than 1 μm2/s.”
Question3: Figure 5: the 3D plot is difficult to understand when printed on 2D paper. Try to improve the results presentation (additional 2D sections?)
Answer: Figures 1 and 2 serve as additional 2D sections derived from the 3D plot illustrated in Figure 5. These figures played a crucial role in identifying HT (Head-Tail binders) outliers and HVO (Highly Viscous) outliers. Apart from the previously identified outliers, new anomalies emerged around kD= 20 mL/g, exhibiting Dnp values higher than anticipated for the observed kD and viscosity. Figure 5 represents the optimal 2D section extracted from the 3D plot, offering a clear depiction of the interrelation between Dnp, kD, and K. This visualization facilitated the establishment of a revised cutoff line (depicted by the black dashed line), supplementing the previously identified cutoff line for HT outliers. With this revised cutoff line, we successfully identified and extracted all outliers from well-behaved molecules, where Dnp, kD, and K exhibit robust correlations.
Response: We have added and highlighted some sentences in main text (also in Figure 5 caption). Figure 5 caption: 3D representation of the data in Figures 1 and 2 showing the nanoparticle diffusion (Dnp), free particle interaction parameter (kD) and dimerization affinity (K) for all molecules in the training set. The faint gray lines represent highly viscous outliers (HVO), the vertical red lines represent previously identified HT molecules in Figures 1 and 2, and green lines represent the remaining molecules in the training set. The red dashed line separates the previously identified HT outliers. The black dashed line is a proposed new cut-off line for the identification of HT molecules.
Question4: Table 1 and 2: the experimental viscosity is given with one significant number after the decimal point, whereas the prediction values are given with two significant numbers. Is this realistic? What is the estimated error of prediction? Is it possible to represent the data in Table 1 and 2 as correlation graph for better assessment? Notably, the experimental viscosity range is much larger (8.1-42.5) compared to the predicted (10-22 for AC-SINS and 10-24 for DLS). As a consequence, the cut-off value for "bad"/"good" molecules (20 cP) is very clear for experimental data but there are quite a few molecules slightly below or above 20 cp in the prediction models and it really depends on the error of prediction (e.g. is 19.95 really below 20 and the molecule is "good"??). What is the reason for the seemingly lower dynamic range of the prediction model?
Answer: We have also kept the predicted values with one significant number after decimal point following the reviewer’s suggestion.
We have added the correlation graph as a better assessment for both validation dataset (Table 1) and test dataset (Table 2) in the supplementary Information.
We've established a binary classification tool with a viscosity cutoff value of 20cP to differentiate between good and bad molecules. Similar binary cutoffs are commonly used in the industry but have an inherent weakness in categorizing borderline molecules.
Since our method is based on a reptation model which assumes the antibody complexes are large enough to behave like an elongated object, yet short enough that it is reasonable to neglect branching, complex fragmentation, and other effects that promote non-reptative relaxation mechanisms, we have seemingly lower dynamic range of the prediction model.
Response:
We have kept predicted values with one significant number after decimal point in Table 1 and Table 2. Also, we have represented the data in Table 1 and Table 2 as correlation graph for better assessment in supplementary Information section mentioning somewhere in main text.
Question5: Comparison with other prediction methods: Did you perform a comparison of the prediction of the here presented model to other models described in the introduction?
Answer: We do not make the claim that our physics-based model performs better than empirical or machine learning methods and, therefore, we have not made the suggested comparison. The advantage of our physics-based approach is providing valuable intuition to guide sequence modification and formulation strategies, as well as improve the model’s performance from hypothesis-driven experiments exploring viscosity and relaxation mechanisms. Despite inherent limitations and assumptions, our approach integrates theory and practical insights. Through this synthesis, we aim to advance our understanding of molecular behavior and contribute to both theoretical and applied progress.
Response: We have included and highlighted this discussion in the conclusion section.
Question6: Application of the prediction: as stated in the conclusion the strength of the model is prediction in the intermediate viscosity regime. What would therefore be the recommendation in terms of selection of an antibody candidate after applying the prediction model?
Answer: We have refrained from making such recommendations directly because it will depend on the characteristics of the pool of candidate molecules. The potential tradeoff between efficacy and viscosity characteristics will likely be determined by how similar the most promising molecules perform.
Question7: Measurements were performed with concentrations of ~70 and ~150 mg/mL. However, the title of the publication draft announces, "prediction of antibody viscosity from dilute solutions". Are there experiments available showing prediction from diluted solutions?
Answer: Actually, the title of our manuscript is “Prediction of Antibody Viscosity from Dilute Solution Measurements" instead of dilute solutions only. Here, dilute solution measurements are and values. These values are obtained from ACSINS and DLS experiments, which are performed in very dilute solutions. Our goal is to predict viscosity at the given concentrations based on the dilute solution measurements.